# System for Visualizing Surface Potential Distribution to Eliminate Electrostatic Charge

**DOI:** 10.3390/s21134397

**Published:** 2021-06-27

**Authors:** Kazuya Kikunaga

**Affiliations:** Sensing System Research Center, National Institute of Advanced Industrial Science and Technology, 807-1 Shuku-Machi, Tosu 841-0052, Saga, Japan; k-kikunaga@aist.go.jp

**Keywords:** surface potential, imaging, electrostatic charge, insulator surface, array sensor

## Abstract

A mixture of positive and negative static charges exists in the same plane on an insulator surface, and this can cause production quality problems at manufacturing sites. This study developed a system with a vibration array sensor to rapidly measure the surface potential distribution of an object in a non-contact and non-destructive manner and with a high spatial resolution of 1 mm. The measurement accuracy differed greatly depending on the scanning speed of the array sensor, and an optimum scanning speed of 10 mm/s enabled rapid measurements (within <3 s) of the surface potential distribution of a charged insulator (area of 30 mm × 30 mm) with an accuracy of 15%. The relationship between charge and dust on the surface was clarified to easily visualize the uneven static charges present on it and thereby eliminate static electricity.

## 1. Introduction

At manufacturing sites, static electricity is generated easily and inevitably because of electrostatic forces and electrostatic discharges. Such electrostatic forces and electrostatic discharges are consequences of static charges that are generated by other physical mechanisms, such as the triboelectric effect. They can then cause major problems such as electrostatic destruction [1,2,3,4], malfunctions of electronic elements (e.g., integrated circuits) and the electronic systems in which they are mounted, and disasters such as explosions and fires [5,6,7]. When two different neutral materials are brought into contact, charge transfer occurs at the interface to equilibrate the energy level and biases the polarity of the material to generate static charges [8]. Further, the generation of static electricity is influenced by and varies with environmental factors like humidity. A small amount of charge has only a small effect on a manufacturing process. However, a large amount of charge and the correspondingly large electrostatic force can produce a mechanical action that can cause a process abnormality. Therefore, the charge must be measured for preventing process abnormalities and thereby reducing production defects.

Thus far, studies have developed noncontact static electricity measurement methods such as electrostatic voltmeters with various types of dielectric electrodes or field-mill type voltmeters [9,10,11,12], microelectromechanical-system-based sensors [13,14], and sound-wave-based methods [15,16,17]. In an insulator, the charge generated by contact or peeling is held where it is generated without diffusing owing to the high surface resistance of the insulating material. As a result, the insulator surface contains a mixture of different amounts of positive and negative charges in the same plane. Therefore, the amount of charge and that left after removing static electricity cannot be evaluated perfectly at only one point. In addition, it is necessary to analyze various situations in these measurements [18,19,20,21]. Therefore, a two-dimensional visualization and quantitative measurement of the charge could enable efficient and effective countermeasures to be taken. Toward this end, point-by-point scanning [22,23] and scanning probe microscope [24,25,26,27] methods have been developed to measure the charge distribution; however, these are time-consuming. To overcome this problem, surface potential sensors have been arranged in an array to shorten the time required for measuring the charge distribution [28]. However, it is technically difficult to miniaturize conventional sensors, and a spatial resolution of 5 mm or less has remained unachievable. To measure the charge distribution, rapid measurements with high spatial resolution must be achieved.

We have previously developed a system consisting of a line-type array sensor, a vibration generator, and a scanning mechanism to process the signals of each sensor simultaneously [29,30]. In the present study, we develop a system that measures the charge distribution rapidly with high spatial resolution by optimizing the scanning speed of the array sensor in the uniaxial direction. This is a new system with a vibration array sensor to rapidly measure (within a few seconds) the surface potential distribution of an object in a non-contact and non-destructive manner and with a high spatial resolution of 1 mm.

## 2. Methods

### 2.1. Theory

A surface potential sensor measures the charge on an object’s surface. In this study, a vibration-type array sensor was used for measuring the surface potential [29]. When the detection electrode and the object’s surface to be measured are installed facing each other, a capacitance *C* (F) is generated between them as an impedance. Here, a charge is assumed to exist on the object’s surface. By vibrating the detection unit side, a spatial displacement is generated, and an AC-modulated signal is induced in the electrodes of the detection unit. The charge *Q* (C) is calculated as
*Q* = Δ*VεS*/(*d* ± *x*),(1)
where ∆*V* (V) is the potential difference between the maximum and the minimum voltages detected by a sensor; *ε* (F/m), the permittivity in air; *S* (m^2^), the area of the sensor; *d* (m), the distance between the sensor and the object; and *x* (m), the amplitude of vibration of the sensor [30]. Here, when the charge *Q*, permittivity *ε*, area *S*, and distance *d* are constant, the potential difference ∆*V* is given by
Δ*V* = *a* ± *bx*,(2)
where *a* and *b* are constant. As shown in Equation (2), the potential difference ∆*V* (V) is represented as a function of only the amplitude *x*. Therefore, the charge *Q* can be measured from these relationships. In this study, the surface potential, which is proportional to the charge, was used to represent the magnitude of the static electricity.

### 2.2. Experimental Setup

Figure 1 shows the experimental system used for measuring the static charge distribution. This system consists of a linear array sensor (self-made printed circuit board), a vibration generator (WaveMaker Mobile, Asahi Seisakusyo Co., Ltd., Tokyo, Japan), a multichannel lock-in amplifier (7210, Signal Recovery, AMETEK Advanced Measurement Technology, Inc, Oak Ridge, TN, USA), and an automatic positioning stage (OSMS20-85(X), Sigma Koki Co., Ltd., Tokyo, Japan). The linear array sensor consists of 30 single parallel electrodes (size: 0.7 mm × 0.7 mm) placed at 1 mm intervals and a vibrating sensor substrate. The material of the sensor (flat-plate electrode) (s_1_ to s_30_) working as a probe is copper. To avoid mutual interference between the flat plate electrodes, the electrodes were designed to be 0.7 mm x 0.7 mm in size and the shielded wire was placed around the electrodes. The array sensor and vibration generator were directly connected and vibrated within the amplitude range of 0.01–1 mm and the frequency range of 0.01–1 kHz. Each sensor was connected directly to the lock-in amplifier from a plate electrode. These sensors were not biased to any voltage, and the alternating current voltage and phases detected by the individual sensors were measured simultaneously by means of the individual interlock amplifier [30].

Three test samples (1, 2, and 3) were used to evaluate the performance of this system. Test sample 1 was a flat plate electrode capable of applying a DC voltage in the range of 0 to ±100 V, as shown in Figure 2a. Test sample 2 was an electrode having a comb-shaped structure, as shown in Figure 2b. The electrodes were placed at 1 mm intervals, electrode width was 0.8 mm, and gap was 0.2 mm. A DC voltage in the range of 0 to ±100 V was applied to each plate electrode (AV_b1_ and AV_b2_) independently. Test sample 3 was a structure in which electrodes having a width of 1 mm were arranged vertically and obliquely, as shown in Figure 2c. These test samples were installed in parallel 0.1–2 mm from and facing the array sensor. The array sensor was fixed to the automatic stage and moved at a scanning speed of 1–20 mm/s while the distance between the array sensor and the sample was kept constant.

## 3. Results and Discussion

### 3.1. Relationship between Signals of Sensors and Surface Potential of Samples

Surface potential distribution of test sample 1 after biasing the sample to 100, 0, and −100 V was measured by a Kelvin-type probe (Trek 6000B-8, Advanced Energy Industries Inc., Fort Collins, CO, USA) linked to an electrostatic voltmeter (Trek 344 Model, Advanced Energy Industries Inc., Fort Collins, CO, USA), as shown in Figure 3a–c. The distance between the probe and the sample surface was 3 mm. These results show that the in-plane surface potential of sample 1 is uniform. First, using such the sample, the array sensor in this study was evaluated. The sensor signal is used to obtain information about the charge of an object’s surface. With the experimental system shown in Figure 1, the linear array sensor substrate was vibrated at a frequency of 200 Hz and an amplitude of 0.25 mm, which was peak to peak. This means that the distance between the sample and the sensors varied from 0.375 to 0.625 mm. Sample 1 connected to the DC voltage source was installed at a distance of 0.5 mm in parallel with the linear array sensor. When DC voltages in the range of 0 to ±100 V were applied to test sample 1, the AC voltage detected by each sensor (s_1_–s_30_) was measured using the lock-in amplifier. Here, the linear array sensor was fixed on the surface to get the potential of a certain location, and the reference signal of the lock-in amplifier was used as a signal of the vibration generator. Figure 3d,e show the relationship between the applied voltages of test sample 1 in the range of 0 to +100 V and −100 to 0 V, respectively, and the voltage detected using the sensor. The voltage detected using the sensor was proportional to the positive and negative applied voltages of 0 to +100 V and −100 to 0 V, respectively. Here, the amplitude of the AC voltage detected using the sensor and the output voltage of the lock-in amplifier are the same. This indicates that the surface potential of the sample and the potential difference detected using the sensor are proportional, as given by Equation (2). The system could thus be used to estimate the surface potential of the sample with an accuracy of 6% at full scale. However, because each sensor has different characteristics, it is necessary to calibrate each sensor.

The potential difference detected using this sensor does not contain information on the positive and negative electrical polarities of the surface potential; however, the phase signal does [30]. Figure 4a,b show the relationship between the voltage applied to test sample 1 in the range of −100 V to +100 V and the phase of the voltage detected using the sensors (s_1_–s_30_). When the absolute value of the applied voltage was ≥20, the phases detected using the sensor were constant at approximately 264° and 84°, respectively. Further, the phases had a relationship with the electrical polarity but not with the magnitude of the voltage applied to the sample. This is because the charge polarity induced by the electrodes on the sensor surface was related, and the phase difference between positive and negative electric fields was 180°. By contrast, the phase was not constant when the absolute value of the applied voltage was ≤20. This is because the signal detected using the sensor was small. These results show that the electrical polarity of the charge on the object can be determined by measuring the phase of the voltage detected using the sensor. In this system, the surface potential and electrical polarity of the object could be measured with an accuracy of 5% in the range of 10 V to 1.5 kV. Further, the array sensor used was clarified to have a detection capability with a spatial resolution of approximately 1 mm [29].

So far, the relationship between the detected voltage and the phase has been shown when the sensor is fixed on a certain point. The purpose of this study was to measure on the sample surface while scanning the sensor. Figure 5 shows the relationship between the surface potential and measured time while scanning the sensor s_10_ at 1 mm/s using sample 1 applied to 100 V. Here, the sensor substrate was vibrated at a frequency of 200 Hz and an amplitude of 0.25 mm and the surface potential was used as a calibrated value, taking one data point in 1 s. The maximum and the minimum of the surface potential were 103.5 and 98.5 V, and the maximum error was 3.5%. Therefore, it was found that the measurement could be performed very stably even while scanning the sensor.

### 3.2. Scanning Speed and Accuracy of a Sensor

To measure the charge distribution with high accuracy by scanning this array sensor, it is necessary to investigate the relationship between the scanning speed of the sensor and the measurement accuracy. For test samples 1 and 2, the change in the signal detected using the sensor was investigated when the scanning speed of sensor s_10_ in the X direction was changed in the range of 1–20 mm/s. Here, the sensor was vibrated at a frequency of 200 Hz and an amplitude of 0.25 mm when the sensor was scanning over the bias sample 2. Figure 6a,b show the changes in the output voltage detected using the sensor through the lock-in amplifier when the applied voltages AV_1_ and AV_2_ were 100 and 0 V, respectively (pattern 1). Here, the time constant of the lock-in amplifier was 10 ms, and the output voltage detected using the sensor was measured at 0.1 mm intervals. The time constant of the lock-in amplifier is the integration time of the signal, and it depends on the time until the detected voltage stabilizes and on the accuracy. When the linear array sensor was fixed on the surface to get the potential of a certain location and the time constant of the lock-in amplifier was 10 ms, it took 40 ms for the detected value to stabilize and the accuracy was 4%. In Figure 6a, peaks and troughs of the detected voltage were observed at the positions centred on AV_1_ and AV_2_, respectively. Specifically, 10 peaks and troughs were found between 5 and 25 mm. This indicates a high (100 V) or low (0 V) surface potential at the peak or trough position, respectively, of the detected voltage. By contrast, the phase of the voltage detected using the sensor was almost constant at approximately 264° in Figure 6b. This is because a positive voltage was applied to test sample 2. Moreover, the phase near the trough of the detected voltage was not constant. This is because of the small detection signal of the sensor, as explained in Figure 4.

Figure 6c,d show the changes in the voltage and phases detected using the sensor at each position when the applied voltages AV_1_ and AV_2_ were +100 and −100 V, respectively (pattern 2). In Figure 6c, peaks and troughs were observed at positions centred on AV_1_ = 100 V/AV_2_ = −100 V and between AV_1_ = 100 V/AV_2_ = −100 V, respectively. Specifically, 20 peaks were observed between 5 and 25 mm. In Figure 6d, the phases at positions AV_1_ and AV_2_ were approximately 264° and 84°, respectively. These results can be explained by the magnitude of the surface potential and the electrical polarity at each position, as shown in Figure 3 and Figure 4. In Figure 6c, The peaks indicate the locations of +100 and −100 V applied to test sample 2. The middle between +100 V and −100 V is 0 V considering the voltage gradient. Therefore, there are 20 peaks and troughs.

However, the detected voltage of each peak was found to change with the scanning speed. Therefore, the error V_err_ is calculated as (V_0_ − V_ave_)/V_0_ × 100. Here, the voltage detected using sensor s_10_ is V_0_ when the voltage applied to sample 1 is 100 V, as shown in Figure 3, and the average value of the detected voltage of each peak between 5 and 25 mm in Figure 6a and 6c is V_ave_. Figure 7a shows the relationship between the detected error and the scanning speed of sensor s_10_ in patterns 1 and 2. The errors for patterns 1 and 2 were slightly lower for scanning speeds of 1–10 mm/s; however, the difference between the two was very small. By contrast, the error for patterns 1 and 2 increased significantly at a scanning speed of ≥15 mm/s, and the difference between the two became large. This indicates that the signal detected using the lock-in amplifier cannot keep up with the change in the signal owing to the movement of the position. When the scanning speed was 10 mm/s, the measurement time per point was 10 ms. It takes 1/4 of the time to fully stabilize. Therefore, when the sensor was scanned, the accuracy was lower than when the sensor was fixed. Figure 7b shows the relationship between the surface potential detected by sensor s_10_ and the scanning time in patterns 1 and 2. Here, the surface potential was a calibrated value. When the measurement time was 3 s (scanning speed was 10 mm/s), the peak surface potential was about 85 V in patterns 1 and 2. Also, when the measurement time was 6 s (scanning speed was 5 mm/s) or more, the surface potential of the peak was about 90 V. These results indicate that a spatial resolution of 1 mm and a measurement accuracy of 15% or less can be achieved at a scanning speed of ≤10 mm/s. In this study, therefore, a scanning speed of 10 mm/s was adopted for rapidly measuring the static charge distribution.

### 3.3. Validity of Surface Potential Distribution

The developed system was evaluated using test sample 3, as shown in Figure 8a. DC 100 V was applied to the metal part of test sample 3. Each sensor was calibrated for its surface potential using test sample 1, the surface potential was measured using each sensor at 1 mm intervals while scanning the sensors, and the surface potential distribution was displayed from their spatial coordinates. Figure 8b shows the measurement results of the surface potential of test sample 3 (measurement area: 30 mm × 30 mm) at a scanning speed of 10 mm/s. The measured surface potential distribution was similar to that of test sample 3, although the diagonal electrode lines formed at 84° and 87° seemed slightly blurred. This means that the measurement result was correct and that we succeeded in visualizing the surface potential distribution with a spatial resolution of 1 mm and an area of 30 mm × 30 mm in 3 s. In this system, the sensor was vibrating during scanning and the distance between the sensor to the surface also changed in the Z direction. This means the area on the sample seen by the sensor was variable. In other words, the detected signal was an average of those areas. This variable distance could influence the data received by the sensor and recorded by the lock-in amplifier. Also, there is the possibility of superposition of one data point to an adjacent data point as well as the influence of the phase shift, but these would be small in the case of a spatial resolution of 1 mm. Additionally, if the sample being tested is not parallel to the detection probe, the accuracy and resolution will vary due to the different areas of the sample seen by the sensor. Therefore, it is necessary to install the sample and the array sensor in parallel. To achieve higher spatial resolution of the surface potential distribution and faster measurement time, increasing the vibration frequency of the array sensor and miniaturizing the size of the detection electrode of the sensor seemed effective in this system. It has also been reported that the resolution can be increased from experiments and simulations by using the characteristics of the electrostatic induction probe of the electrostatic voltmeter [31,32]. Further high spatial resolution can be expected by combining the surface potential distribution obtained in this study with such a simulation.

### 3.4. Visualization of Contact Charging Distribution

To evaluate the developed system, vinyl chloride was contacted with glass to generate charge and thereby serve as a charged sample. Figure 9a shows the measurement results of the charged glass after vinyl chloride was brought into contact with it along the letters ‘AIST’. The measurement time was 6 s, measurement area was 30 mm × 60 mm, and scanning speed was 10 mm/s. A visualization of the surface potential distribution of the charged glass indicated that the contacted position was charged positively. This was explained by the triboelectric series, in which glass and vinyl chloride are positively and negatively charged, respectively. Next, the charged glass was exposed to an ionic wind generated using an ionizer. The surface potential measurement result (Figure 9b) indicates that after using the ionizer, the charged letters ‘AIST’ disappeared, that is, the electric charge was removed. By using the developed system in this manner, it was possible to easily evaluate the position and amount of charge generated by contact as well as the static elimination effect.

### 3.5. Distribution of Charge and Dust

Finally, the relationship between charge and dust was investigated. Charge was generated when acrylic and vinyl chloride were brought into contact along the letter ‘X’, as shown in the measurement results of the surface potential distribution in Figure 10a. Next, after putting the cotton dust from the fibers on the charged acrylic, dust was sprayed using an air duster on the acrylic surface. At that time, the dust was seen to remain attached to the charged areas, as shown in Figure 10b. Further, this dust could not be removed completely when the charged acrylic was exposed to an ionic wind generated using an ionizer. This indicates that dust easily adhered to the charged place, following which it was difficult to remove. In other words, properly measuring and eliminating static charge in advance at a manufacturing site can help prevent dust adhesion.

## 4. Conclusions

In this study, we constructed a system that scans the surface potential of an object in a non-contact and non-destructive manner by using a line-type array of 30 sensors. This system could measure the surface potential stably with a measurement accuracy of 15% when the scanning speed of the array sensor was set to 10 mm/s. In addition, the surface potential of an area of 30 mm × 30 mm could be measured rapidly with a high spatial resolution of 1 mm within 3 s. In this paper, we verified the spatial resolution of 1 mm as a target. There is a possibility that the spatial resolution can be further increased by the scaling law by downsizing the array sensor in this system. Further, the relationship between charge and dust was verified. The results indicate that the proposed system can be used to eliminate static electricity and thereby improve production quality at manufacturing sites. To verify the surface potential, distribution with high spatial resolution and evaluation of the change during the time of the distribution are required in the field of next-generation electronics such as flexible printed electronics and power semiconductors. The surface potential distribution can be measured in detail in a few seconds in this system. This means that the time variation of the distribution can be verified. It can also be used as an evaluation or analysis tool for the mass production of high-quality, high-value-added products.

## Figures and Tables

**Figure 1 sensors-21-04397-f001:**
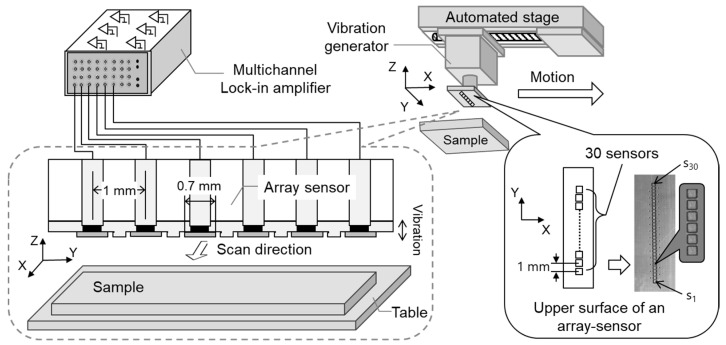
Experimental system for measuring static charge distribution.

**Figure 2 sensors-21-04397-f002:**
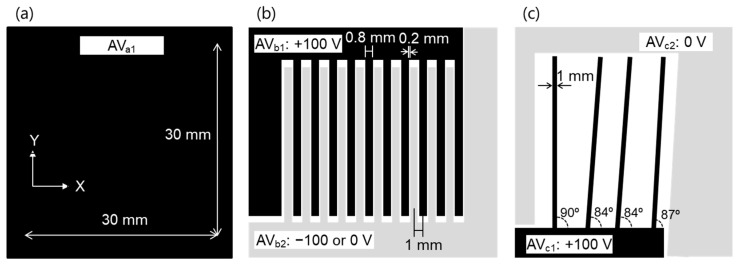
Shape of (**a**) test sample 1, (**b**) test sample 2, and (**c**) test sample 3.

**Figure 3 sensors-21-04397-f003:**
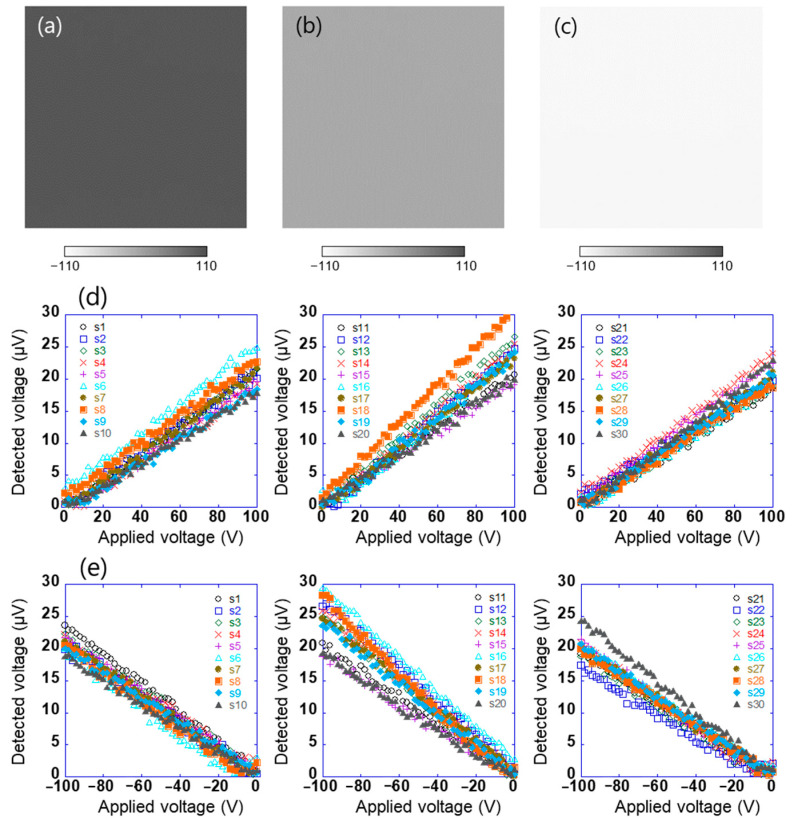
Surface potential distribution of test sample 1 which was applied (**a**) 100 V, (**b**) 0 V, and (**c**) −100 V, and the relationship between applied voltages of test sample 1 in the range of (**d**) 0 to +100 V and (**e**) −100 to 0 V and voltage detected using each sensor (s_1_–s_30_).

**Figure 4 sensors-21-04397-f004:**
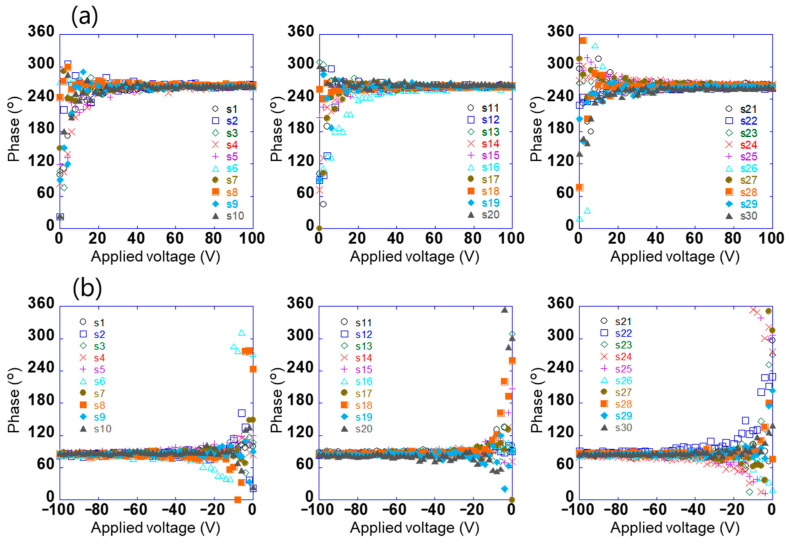
Relationship between voltage applied to test sample 1 in the range of (**a**) 0 to +100 V and (**b**) −100 to 0 V and the phase of the voltage detected using each sensor (s_1_–s_30_).

**Figure 5 sensors-21-04397-f005:**
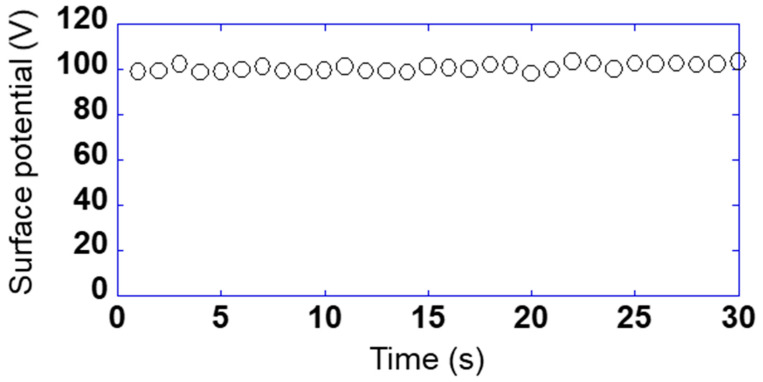
Relationship between the surface potential and measured time while scanning the sensor s_10_ at 1 mm/s using sample 1 applied to 100 V.

**Figure 6 sensors-21-04397-f006:**
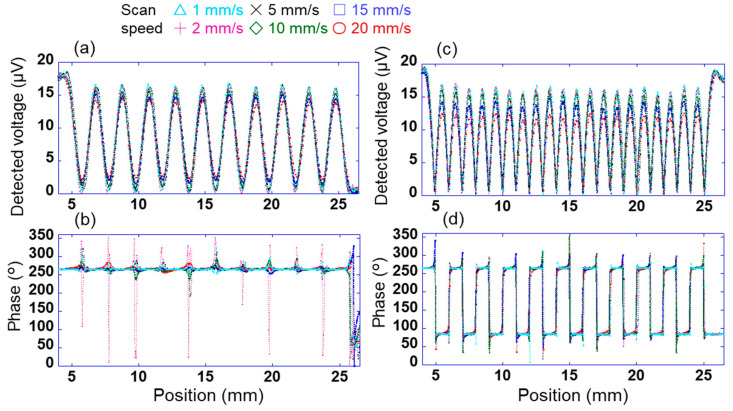
Changes in signal detected using sensor s10 through the lock-in amplifier with scanning speeds of 1–20 mm/s. Changes in voltage detected using the sensor when the applied voltages to sample 2 are (**a**) AV_1_ = 100 V and AV_2_ = 0 V (pattern 1) and (**c**) AV_1_ = 100 V and AV_2_ = −100 V (pattern 2). Changes in phases detected using the sensor when the applied voltages to sample 2 are (**b**) pattern 1 and (**d**) pattern 2.

**Figure 7 sensors-21-04397-f007:**
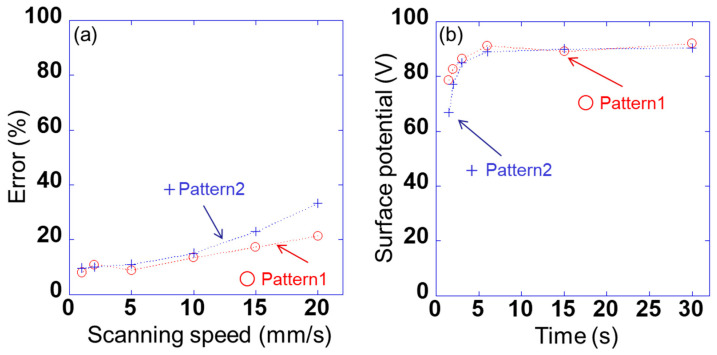
(**a**) Relationship between detected errors V_err_ and scanning speed of sensor s_10_ in pattern 1 (AV_b1_ = +100 V and AV_b2_ = 0 V) and pattern 2 (AV_b1_ = +100 V and AV_b2_ = −100 V) and (**b**) the relationship between the surface potential detected by sensor s_10_ and the scanning time in patterns 1 and 2.

**Figure 8 sensors-21-04397-f008:**
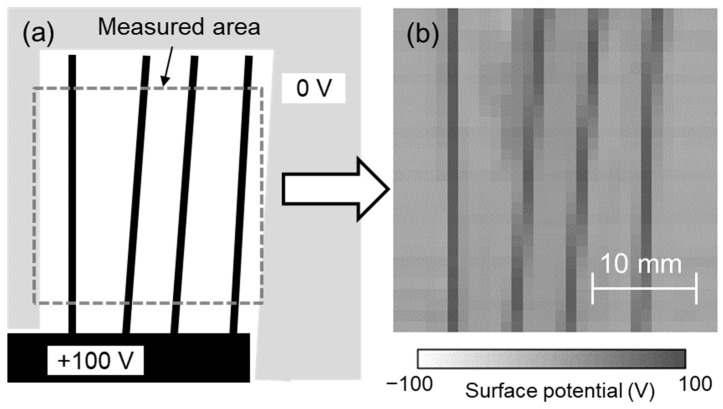
(**a**) Test sample 3 and (**b**) surface potential distribution measured using developed system.

**Figure 9 sensors-21-04397-f009:**
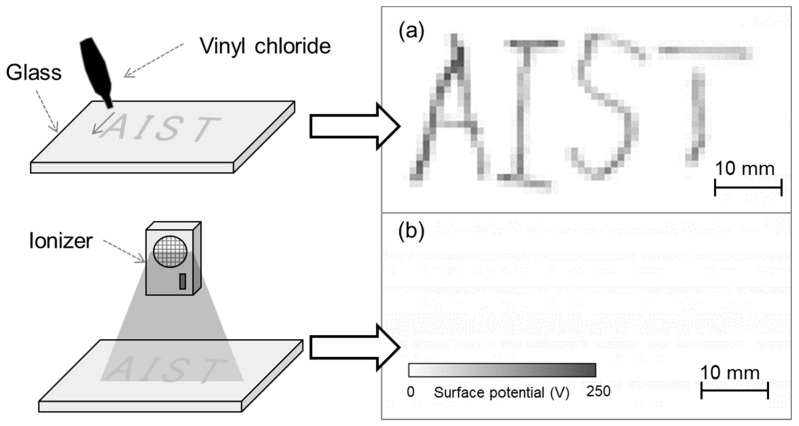
(**a**) Surface potential distribution of charged glass upon contacting vinyl chloride and (**b**) surface potential distribution after eliminating static.

**Figure 10 sensors-21-04397-f010:**
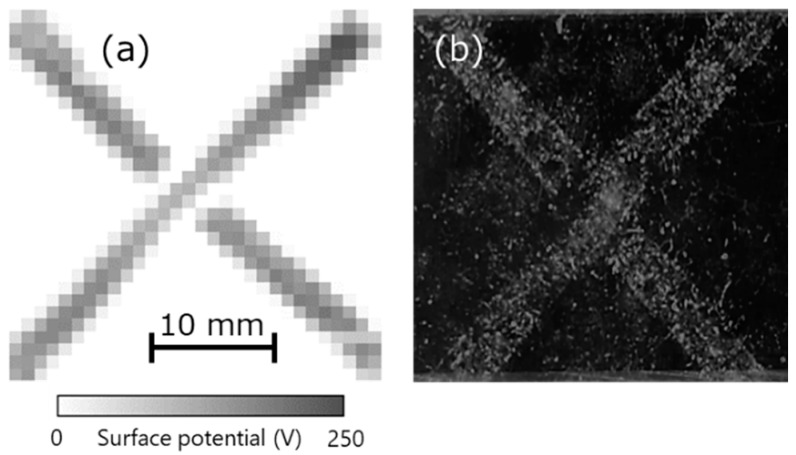
(**a**) Surface potential distribution of charged acrylic upon contacting vinyl chloride and (**b**) photograph after attaching dust.

## Data Availability

Data sharing not applicable.

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
