# Peer review of "System for Visualizing Surface Potential Distribution to Eliminate Electrostatic Charge"

_sensors, 2021, doi:10.3390/s21134397_

Round 1

Reviewer 1 Report

This is an excellent experimental finding. I appreciate the author working on this issue. I like to include some of my concerns. I hope the author will be happy to answer.

Q1. What is the material of the sensor (flat-plate electrode) (s1 to s 30) working as a probe? Is this sensor biased to any voltage before reading the surface potential? Is there any influence of this sensor on the static charge of the surface (surface voltage) that influences the reading (voltage) recorded by the Lock-in amplifier? Is it possible to add the schematic of the self-made circuit diagram?

Q2. In Figure-1, is every single sensor connected to the individual channel of the lock-in amplifier? This figure-1 does not represent this. It is understood that each sensor is connected to a common wire and this common wire is connected to each channel of the lock-in amplifier.

Q3. This point is to get a clear picture of my understanding. The sample surface to probe (sensor) surface distance was 0.5 mm. When the sensor was vibrating with an amplitude of 0.25 mm, distance (sample to the sensor) varied from 0.25 mm to 0.75 mm. Am I correctly understood? If I am correct. I would recommend adding this sentence on page-3 in lines 111-112.

Q4. I would recommend adding surface potential (2D distribution, like Fig. 7,8, and 9) data after biasing sample-1 to 100 V, 0 V, and -100V, taken by another commercially available non-contact voltmeter. I am sure, this will strengthen this experimental finding.

Q5. About sample-1 data (Fig. 3): During the data collection, was the linear array sensor scanning the sample surface (along the x-axis), or was it fixed on the surface to get the potential of a certain location?

If it was fixed on a certain point, only one data point was taken from the biased sample by a specific sensor. In this case, data precision was low. If the surface had been scanned along the x-axis, data precision was high. I would recommend clarifying this.

Q6. If the surface has been scanned, I strongly recommend adding an extra figure of surface potential (recorded by the Lock-in Amplifier) VS time, keeping the sample-1 potential fixed. I am sure, it will increase the reliability of the measurement that might shade brighter light on the phase shift effect. If it was not done, I strongly advise the author to do it and resubmit.

Q7. About sample-2 data (Fig. 4): During the scanning along the x-axis, it was mentioned that the lock-in amplifier’s time constant was 10 ms. What was the role of this number on the data? Please clarify this. (Page-5, line: 158)

Q8. Output voltage detected using the sensor was measured at every 0.1 mm interval. (Page-5, line: 159). Was it the spatial resolution? If yes, to scan 1 mm distance, the sensor was taking 10 data points. If the scanning speed was 1 mm/s, it was taking 10 data points in 1 second. Was this spatial resolution constant throughout the whole experiment? What was the role of time-constant of lock-in amplifier in this situation? Would you please clarify this?

Q9. About sample-2 data (Fig. 5): Sensor was scanning two comb-like surfaces (electrically isolated) facing each other, where every adjacent tooth had been biased differently. I strongly recommend adding a plot (surface voltage, detected by a lock-in amplifier, VS time). This will clarify the performance of the sensors in a significantly better way.

Q10. About sample-2 data (Fig. 5a): There are 10 teeth (Fig. 2, sample-2, if the surface is like a comb). These teeth were biased to 100 V and 0 V and scanned at different speeds. There are 10 peaks (10 troughs) of voltage detected by the sensor and lock-in amplifier, shown in Fig. 5a.  The same sample (biased to 100 V and -100 V, Fig. 5c), shows 20 peaks (20 troughs). Why it was? Was it due to the lock-in amplifier? It should be clarified.

Q11. While the sensor was scanning over the bias sample-2, was the sensor vibrating, like the case on sample-1? It needs to be clarified.

Q12. If it was vibrating during scanning, the distance between the sensor to the surface also changing in the z-direction. In this situation, was the area on the sample seen by the sensor/probe (0.7 mm X 0.7 mm) constant or variable? Was there any influence of this variable distance (z-axis) on the data received by the sensor and recorded by the lock-in amplifier?

Q13. Is there any possibility of superposition of one data point to an adjacent data point? Does it influence the phase shift?

Reviewer 2 Report

This paper presented an interesting method to image surface potential distribution of non-conducting object. Overall it is a well-written paper and the method is of novelty. However, I can not recommend its current form for publication. Please address my following concerns:

  1. Please provide a more detailed description on the experimental setup.
  2. Line 188 discusses the results of Figure 6, but the text analysis and the figure does not correspond, please double check.
  3. In line 191, the author mentioned "These results are presented a spatial resolution of 1 mm and a measurement accuracy of 15% or less can be achieved at a scanning speed of ≤10 mm/s. In this study, therefore, a scanning speed of 10 mm/s was adopted for rapidly measuring the static charge distribution.” The scanning speed of 10mm/s is then continued for experimental verification, however, this speed is obtained by the two specific electric field distribution situations generated by test block 2, when the charge in the application does not meet the above two specific conditions, can the accuracy of 15%, 1mm resolution effect? Is the generality of this resolution calibration method feasible?
  4. Please comment on what if the sample being tested is not parallel to the detection probe.

Reviewer 3 Report

The paper reports about “System for visualizing surface potential distribution to eliminate electrostatic charge”. The topic of the reviewed article is a very interesting for technical applications. However, the manuscript requires a revision prior to publication.

The following Suggestions and Comments have to be addressed before publication of the paper:

1.In Introduction, it is insufficiently presented a novelty or an innovation of the manuscript. Please introduce what is novelty or innovation in this article. Also, please indicate an aim of developed system.

2.In Figures 5a-d, please change location of designations ‘a, b, c, d’, it will be better solution that these designations do not locate in the area of graphs.

3.In Figure 6, the quality of presented characteristics is a poor, please correct this.

4.In Conclusions, please indicate further experimental research in this scientific field, what requires and needs a wider investigation and development.

5.This manuscript has one Author, but in ‘Conflicts of Interest’ is indicated few Authors. Please check this.

Round 2

Reviewer 1 Report

Thank you very much for revising this article. I appreciate the appropriate response to the points I mentioned early. After double-checking the English language, I hope the editor will be happy to accept it for publication. 

Reviewer 2 Report

the authors have addressed previous concerns and comments. I am happy to recommend the paper for publication.